# Psychological and Medical Characteristics Associated with Non-Adherence to Prescribed Daily Inhaled Corticosteroid

**DOI:** 10.3390/jpm10030126

**Published:** 2020-09-14

**Authors:** Brett G. Toelle, Guy B. Marks, Stewart M. Dunn

**Affiliations:** 1Woolcock Institute of Medical Research, The University of Sydney, Sydney 2006, Australia; g.marks@unsw.edu.au; 2Sydney Local Health District, Sydney 2050, Australia; 3South Western Sydney Clinical School, University of New South Wales, Sydney 2170, Australia; 4Ingham Institute of Applied Medical Research, Sydney 2170, Australia; 5Department of Psychological Medicine, The University of Sydney, Sydney 2006, Australia; stewart.dunn@sydney.edu.au

**Keywords:** adherence, asthma, inhaler use, personality, anxiety, depression, optimism, knowledge, communication

## Abstract

Medication non-adherence is associated with sub-optimal asthma control. Identification of medical and psychological characteristics associated with non-adherence is important to enable a targeted and personalized approach when working with patients and for the development of interventions to improve patient outcomes by improving medication adherence. We enrolled adults who had diagnosed asthma and who were prescribed daily inhaled corticosteroid medication. We used published and validated instruments to measure medical characteristics including asthma features, practical asthma knowledge and perceived involvement in care and psychological characteristics including anxiety, depression, optimism, and personality traits, to assess the relationship with medication non-adherence. A total of 126 participants provided data, with 64 (50.8%) of the participants identified as non-adherent. Multivariate analyses showed that younger age, high neuroticism scores and a previous asthma hospital admission were associated with non-adherence. Interestingly, depression was associated with a lower risk of non-adherence. This study showed that a personalized medicine approach would include interventions targeting those who are younger, who have been in hospital for asthma and who rate high on the neuroticism personality trait. Given the availability of effective medications for asthma, better understanding of the characteristics associated with non-adherence is important to enhance optimal self-management.

## 1. Introduction

Asthma continues to cause significant morbidity in the community [1] despite wide dissemination of asthma management plans and the availability of effective anti-inflammatory drugs [2,3]. One of the reasons for the continued high level of morbidity is that adherence to prescribed asthma medication is often sub-optimal, with studies reporting a range of between 30% to 70% adherence [4,5,6]. Studies investigating the factors that influence adherence to prescribed medication are important if we are to reduce the morbidity associated with asthma. Therefore, the identification of factors that influence medication adherence has the potential to lead to the development of more effective, personalized and targeted intervention strategies to enhance adherence and lead to improved outcomes.

Non-adherence has been recognised as a significant issue by a number of health organisations. In particular, the Australian National Asthma Council [7], the Royal Pharmaceutical Society of Great Britain [8] and the World Health Organization [9] have published seminal guidelines to assist health professionals to address the issue of non-adherence. Yet, non-adherence with medication remains a significant barrier to optimal asthma control and is as relevant today as it was when these guidelines were published two decades ago.

The World Health Report “Adherence to Long-Term Therapies: Evidence for action” describes the complexity of adherence and proposes five dimensions of adherence [9]. These dimensions include Social/Economic factors, Therapy-related factors, Patient-related factors, Condition-related factors and Health system/Health care team-related factors. These generic factors apply in different ways to different health conditions but provide a useful framework in which to view this complex area.

In studies of non-adherence in asthma most studies have investigated social and economic factors, therapy factors and condition related factors [Ref review articles]. These studies have reported that cost of medications, complexity of the therapeutic regimen, and severity of asthma to name just a few were associated with medication non-adherence. However, in this study we aimed to add to the knowledge base in asthma by investigating patient-related and health care team-related factors that had not previously been well studied in asthma or studied at all.

For example, the patient-related factors of personality had been shown to be associated with non-adherence in renal dialysis [10] and diabetes [11] but we were unaware of reports in asthma. Additionally, optimism had been investigated as a mediator of adherence in patients undergoing cancer treatment [12] but not in asthma. With regard to health care team-related factors, reports of the importance of the therapeutic relationship, communication skills and shared decision on medication non-adherence [13] lead us to investigate the role of patients perceived involvement in their care.

Therefore, this study aimed to assess medical and psychological factors associated with non-adherence to prescribed daily inhaled corticosteroid medication. The hypothesis we were testing was that practical asthma knowledge, perceived involvement in care, anxiety, depression, optimism and personality traits are associated with non-adherence to prescribed daily inhaled corticosteroid medication use.

## 2. Materials and Methods

### 2.1. Participants

Participants representing a broad range of people with asthma were recruited from two sources: an adult outpatient specialist referral clinic of a large metropolitan teaching hospital and a database of adults who had their name recorded on the research volunteer database at the Woolcock Institute of Medical Research. The clinic patients were asked, at their initial clinic visit, to complete a questionnaire about their medical history and treatment and whether they consented to being contacted about research projects. The research volunteer database contained the names of adults who have participated in previous research studies and who have agreed to be contacted in the future. It also contains the names of adults who have contacted the Institute in response to an advertisement or a media item and who have volunteered to be involved in a research program.

Eligible participants were adults who had diagnosed asthma, who were prescribed daily inhaled corticosteroid (not combination including formoterol) medication, who agreed to participate in research and who were fluent in English. All participants provided informed written consent. The protocol was approved by the University of Sydney Human Ethics Committee, Ref 01/08/44.

### 2.2. Study Design

Potential participants were contacted by telephone or mail and were informed that we were conducting research investigating the way people manage their asthma. Participants were told they would be sent questionnaires and would receive a call in seven days. All participant questions seeking further information or clarification were answered honestly and completely. However, we did not discuss adherence or compliance. Participants who agreed to participate and who met the study criteria were enrolled and mailed the study materials.

Seven days after the participants received the recruitment call we telephoned them to ensure the study materials had arrived, to assist administer one of the questionnaires by interview and finally to ask the participant to return the questionnaire booklets.

### 2.3. Materials

#### 2.3.1. Inhaler Adherence scale

For this study, the 4 item Inhaler Adherence Scale (IAS) [14] was used to measure inhaler adherence. This scale asks simple questions about the participants’ inhaled medication use over the previous 3 months. The four areas that are included are taking medication less than prescribed due to forgetting, being careless or feeling better and stopping medication due to feeling better. The scale has the advantage that it is generalizable to any corticosteroid inhaler and applicable whether it was a pressurised metered dose inhaler (pMDI) or dry powdered inhaler (DPI). The IAS is a simple four -item ‘yes’/‘no’ questionnaire where each ‘yes’ response was scored as 0 and each ‘no’ response was scored 1. The sum of the scores adds to a possible score of between 0 and 4 with a higher score indicating greater adherence. In our electronic validation study, IAS scores of ≤ 2 had a sensitivity of 73% and a specificity of 80% for detecting non-adherence. The area under the ROC curve was 0.764 (*p* < 0.001) [15].

#### 2.3.2. Demographic and Asthma Characteristics

All participants provided general information about age, gender, possession of an asthma action plan, past participation in asthma education, previous hospital admissions, age of diagnosis and frequency of asthma symptoms.

#### 2.3.3. Anxiety and Depression

The Hospital Anxiety and Depression Scale (HADS) was used to measure anxiety and depression [16]. The HADS is a 14-item questionnaire with seven anxiety and seven depression questions, each with four response options about the frequency of feelings, ranging from most of the time, some of the time, a little of the time to not at all. Scores for anxiety and depression scaled range from 0 to 21. A score of less than 8 for either state is classified as normal, between 8 and 11 as mild and 12 or more as moderate to severe [17].

#### 2.3.4. Personality

The Neuroticism Extraversion Openness Five Factor Inventory (NEO-FFI) developed by McCrae and Costa [18], was used to measure the big five domains of personality, which are neuroticism, extroversion, openness, agreeableness and conscientiousness. The NEO-FFI is a shortened version of the longer NEO-PI questionnaire. The NEO-FFI was provided as a separate 60-item questionnaire booklet with each personality domain assessed by 12 facet questions. For each question there were five response items from ‘strongly agree’ to ‘strongly disagree’. The NEO-FFI took approximately 15 min to complete and provided similar information to the longer 240-item NEO-PI. The five domains were classified low, medium or high according to population norms. In 2004, the authors undertook a review of items included in the original instrument and found that although some items could be updated, this resulted in only modest changes in reliability and factor structure with no change in validity. Therefore, they recommended that the continued use of the original version was “reasonable for most applications” [19].

#### 2.3.5. Optimism

Optimism was measured using a questionnaire designed by Seligman and published in his book *Learned Helplessness* [20]. This scale is a 48-item questionnaire with two alternative answers for each question. The 48 items provide an overall optimism score where scores of 6 to 8 are classified as moderately optimistic, 3 to 5 average, 1 to 2 moderately pessimistic and 0 or below as very pessimistic. The questions measure the three crucial dimensions of a person’s explanatory style, which are permanence, pervasiveness and personalisation. When explaining a situation or event, an individual chooses options amongst these three crucial dimensions. The permanence dimension can be permanent (pessimistic) or temporary (optimistic), the pervasiveness dimension can be universal (pessimistic) or specific (optimistic) and the personalisation dimension can be external (pessimistic) or internal (optimistic).

#### 2.3.6. Practical Asthma Self-Management Knowledge

Practical asthma self-management knowledge was measured using two specific asthma management scenarios (1) a slow onset asthma attack and (2) a fast onset asthma attack. These hypothetical asthma attack scenarios were described by Sibbald [21] and modified by Kolbe [22]. Each scenario had three stages to indicate progressive deterioration in asthma control. The slow onset attack occurs over a seven-day period and the fast onset asthma attack occurs over an hour. The scenarios were provided to the participants in written form but were also repeated over the telephone to ensure that participants had all the relevant information required to describe their management steps. Individual responses were recorded and scored by the interviewer [BGT] according to the published scoring schedule. Each scenario was scored out of a possible 25 points, with 25 representing optimal management in response to the hypothetical scenarios. Optimal practical self-management was indicated by responses that include appropriate responses to worsening symptoms or peak flow measurements, increasing medication dose and timely seeking of medical aid with ongoing deterioration.

#### 2.3.7. Perceived Involvement with Care

Information about participant perception of their involvement in their asthma care was collected by the Perceived Involvement in Care Scale (PICS) [23]. The PICS is a 13-item ‘agree’/‘disagree’ questionnaire that is simple to understand and requires about three minutes to complete. The scale has three sub-scales: (1) Doctor facilitation of patient involvement, (2) Level of information exchange, and (3) Patient participation in decision-making. A ‘disagree’ response is scored as a value of 1 and an ‘agree’ response is scored as a value of 2. Subscales scores for “Doctor Facilitation” range from 5–10 and for “Information exchange” and “Patient decision-making” range from 4–8.

### 2.4. Sample Size

The sample size for this study was calculated for a separate analyses. However, the sample size of 126 provides at least 80% power (alpha 0.05) to detect an OR of at least 3 based on conservative estimates of a 50% prevalence of non-adherence and a 50% prevalence of exposure.

### 2.5. Statistical Analyses

All data were analysed using the SAS statistical package for Windows [24]. Non-adherence was defined as an IAS score ≤ 2. For risk factors measured on a continuous scale, comparisons between non-adherent and adherent groups was tested using an independent samples t-test for normally distributed variables and a Wilcoxon non-parametric test for non-normally distributed variables. For binary risk factors, associations with non-adherent vs. adherent status was quantified as odds ratios and tested using Chi-square tests with Yates continuity correction. The Cochrane-Armitage trend test was used to test for significant linear association of non-adherence across ordered categorical exposure variables.

Multivariable logistic regression was used to identify independent predictors of non-adherence. All variables that had a *p* value of ≤ 0.1 were included. The final model includes all variables.

## 3. Results

A total of 175 potentially eligible individuals were identified from the two recruitment sources, 145 (83%) adults agreed to participate, 5 (3%) refused and the remaining 25 (14%) were unable to be contacted.

Of the 145 participants who agree to participate and were sent the study materials, 19 (13%) either said that they had in fact mailed back the materials, did not return their materials after multiple contact attempts or actively withdrew from the study. The study sample was balanced with regard to gender and recruitment source (Table 1). Participants had a mean age of 49 years (19 to 86 years) and the median age at asthma diagnosis was 13.5 years. The majority of members of the sample reported having an asthma management plan. Approximately half of the participants (64, 50.8%) had an Inhaler Adherence Score of ≤ 2 and hence were defined as non-adherent.

The proportion of non-adherent participants was not different between those recruited from the Asthma Centre and from the Volunteer database (Table 2). However, younger people were significantly more likely to be non-adherent than older people and male participants were more likely to be non-adherent than females. Most clinical features of asthma, apart from a history of previous hospital admission for asthma, were not significantly associated with non-adherence (Table 3). Among the psychological factors assessed only higher levels of neuroticism was significantly associated with non-adherence (Table 4). Higher levels of anxiety, lower levels of depression, within the personality traits, higher levels of extraversion and openness and lower levels of conscientiousness and agreeableness all tended to be associated with non-adherence, but these were not statistically significant. Other factors investigated such as Perceived Involvement in Care and its sub-scales, the practical knowledge score for fast or slow onset asthma attack and the level of optimism were similar between non-adherent and adherent groups (Table 5).

The significant independent predictors of non-adherence from the final multivariate model are shown in Table 6. Significant independent predictors of non-adherence were lower age, higher neuroticism, the absence of depression, and a history of a previous hospital admission for asthma in the last year.

## 4. Discussion

This study showed that among adults with asthma who were recruited from a hospital and community setting, approximately half of the sample was non-adherent with their prescribed daily inhaled corticosteroid medication. Non-adherence was predicted by younger age, the absence of depression, greater neuroticism and previous asthma hospital admission. However, many factors were not associated with non-adherence. They included having received asthma education, having an asthma management plan, anxiety, the personality domains of extraversion, openness, agreeableness, and conscientiousness, optimism, practical asthma knowledge and perceived involvement in asthma care.

The strengths of this study included the use of published validated questionnaires and high participation and low dropout rate. Adherence was classified by the Inhaler Adherence Scale [14] and a range of independently developed, validated and published instruments were used to measure individual psychological and medical characteristics. This provides confidence that both adherence, psychological and medical characteristics were validly measured and therefore the associations identified are likely to be accurate.

Our study materials referred to our interest in understanding general asthma management so that we did not encourage or discourage any particular type of participant contributing to the study. The high consent rate from the eligible contactable adults (145/150) provided reassurance that there was no responder bias amongst those who agreed to participate in the study. Also, the low drop-out rate (19/145) was additional evidence against responder bias.

The potential limitations of this study include the use of a range of self-report instruments. Non-adherence was assigned based on a simple four item questionnaire that asked about medication taking behaviour in the previous three months. However, our validation of the IAS against electronically monitored inhaler use, previously described, gave us confidence that this instrument was able to identify non-adherence [15]. Electronic monitoring of adherence is not without its own challenges because the attachment of an electronic device to an inhaler can serve as a reminder and influence behaviour. We also restricted participants to those who used an ICS inhaler, not in combination with formoterol, because measuring adherence with a combination inhaler containing formoterol would be influenced by the rapid onset of symptomatic relief provided by the bronchodilator. Additionally, studies with electronic monitoring are typically restricted to one type of inhaler and so the findings are specific to patients prescribed and using that type of inhaler. We did not collect information about asthma control or smoking history so we are not able to examine the association of these characteristics with non-adherence which would have been quite interesting. In this study we did not find an association between daily corticosteroid dose and adherence but a larger study would be sufficiently powered to perform a stratified analyses of the association of these psychological and medical characteristics with adherence for different levels of daily corticosteroid dose.

All medical and psychological characteristics were collected by self-report instruments and it is possible that participants constructed their responses to present themselves in the most positive light. However, we attempted to use instruments that had been developed and published with sufficient supporting data.

It is also important to recognise that these associations do not imply a causal role (or lack of causal role) for the risk factors we tested. The model is a predictive model, designed to identify those at risk of non-adherence but should not be interpreted as a causal model. For example, it is plausible that the association with a history of hospital admissions for asthma is causal in the opposite direction; that is, patients who are non-adherent to inhaled corticosteroid therapy are more likely to have exacerbations requiring hospital admission. 

In this study, neither having received asthma education or possessing an asthma management plan was associated with adherence. Although it is plausible that these factors would demonstrate a relationship with non-adherence, a recent Cochrane systematic review found that the included studies were of low quality and did not demonstrate that having a personalised asthma action plan (PAAP) conferred a health benefit alone or in combination with education [25]. Anxiety was not associated with non-adherence in this study, which concurs with a historic meta-analyses [26] but disagrees with some more recent research [27] demonstrating an association across a range of medical conditions. Interestingly, in a study of adults who had asthma from Nordic countries an association between anxiety and non-adherence was demonstrated for exacerbations but not for normal situations. Our measure of non-adherence did not discriminate between adherence during usual care or during exacerbations.

We have not been able to identify any reports where optimism has been investigated as a factor associated with non-adherence in asthma. However, in a study of patients after an acute coronary syndrome event, optimism two weeks after the event was predictive of self-reported adherence at six months [28]. These authors, suggested that interventions that actively promote positive constructs such as optimism may confer longer term benefit. The role of the positive psychology construct of optimism in non-adherence is still open for investigation.

Personality traits as identified within McCrae and Costa “Big Five Model” of Neuroticism, Extraversion, Openness, Agreeableness and Conscientiousness were investigated in a similar study to this study. Asthma medication non-adherence was measured by questionnaire and the NEO-FFI was used to collect information on personality. In that study higher conscientiousness scores were associated with lower non-adherence [29]. In a study of treatment adherence in cancer outpatients the personality traits of conscientiousness and agreeableness were positively associated with treatment adherence. In another study, conscientiousness was found to be associated with adherence to the oral contraceptive pill but in a multivariate model only accounted for 7.1% of the variance in adherence. In this study we found that neuroticism was associated with non-adherence, which we have not seen previously reported in the adherence literature. 

We explored the role of practical asthma knowledge assessed via responses to asthma exacerbation scenarios to expand our understanding of the role of education as a strategy to enhance adherence. A Cochrane systematic review published in 2002 [30] but not updated since showed that limited (information only) patient education programs for adults who have asthma did not improve health outcomes. However, a more recent review reporting on adults who had received an educational intervention after attendance at an emergency room for asthma showed more that educational interventions reduced further admissions to hospital for these patients [31]. Patient education remains an important component of asthma management promoted by guidelines [2] however there are open questions about when best to deliver this intervention and the circumstances under which patients draw on this knowledge.

Shared decision making has been promoted widely as a means to engage patients in the management of their chronic illnesses. A study of shared decision making in asthma demonstrated significant improvements in adherence and health outcomes [32] and in another study that only included women, benefits were seen for adherence but not for symptoms [33]. The instrument we chose to use to measure this construct was the Perceived Involvement in Care Scale (PICS) and although the scale and sub-scales seemed appropriate for the task, we did not demonstrate any significant associations. 

Numerous reports have shown that medical practitioners have difficulty identifying which of their patients are likely to be non-adherent [34,35,36]. We tested the association of a wide range of psychological and medical characteristics with non-adherence to prescribed daily inhaled corticosteroids including anxiety, optimism, personality or the medical characteristics of perceived involvement in care or asthma practical knowledge and did not find associations with non-adherence. All of the characteristics included could plausibly be associated with non-adherence however only a few were demonstrated to have a statistically significant association. This study demonstrated why trained health professionals have such difficulty.

The finding that moderate to severe depression was associated with better inhaler adherence conflicts with some of the commonly observed features of depression, such as general loss of interest or pleasure, decreased activity, lack of motivation, social withdrawal or feelings of hopelessness [37]. A meta-analyses of studies has shown opposite effects, with higher levels of depression being associated with non-adherence [38]. Potential explanations for the contrasting results may be differences in the populations being studied, the tools used to measure depression and the severity of asthma among participants. One possibility for this unexpected association is that participants who were depressed took their anti-depressant medication and this acted as a prompt to take their ICS medication. Unfortunately, we do not have information about other concomitant medications and so this hypothesis is speculation, but worthy of investigation in future studies. In this study, older age was associated with a decreased risk of non-adherence, which is consistent with the findings from earlier studies [39,40]. But, elderly adults who have asthma face a range of functional, cognitive and medical issues that may lead to a lower level of adherence and for which interventions have been designed specifically to address this challenge [41].

## 5. Conclusions

The finding that in a multivariate model only age, neuroticism, depression, and previous hospital admission for asthma, amongst a range of characteristics, were associated with non-adherence to prescribed daily inhaled corticosteroid medication provides evidence of how difficult it is to predict non-adherence. However, these characteristics are identifiable and offer opportunities for the development of personalized interventions to address non-adherence. Clearly, medication adherence is a complex issue that requires ongoing research so that we can identify factors associated with non-adherence and develop randomised controlled trials of interventions that can be implemented to improve medication adherence and health outcomes.

## Figures and Tables

**Table 1 jpm-10-00126-t001:** Characteristics of the 126 participants who completed the study.

Characteristic	
Male (%)	46.0
Mean age years (SD)	49.0 (15.8)
Median age asthma diagnosis years (IQR)	13.5 (36.0)
Recruited from Asthma Clinic (%)	49.2
Had an asthma management plan (%)	79.4
Prescribed daily inhaled corticosteroid	
Beclomethasone dipropionate (%)	12 (9.5)
Budesonide (%)	21 (16.7)
Fluticasone propionate (%)	37 (29.4)
Fluticasone propionate/salmeterol (%)	56 (44.4)
Daily dose of inhaled corticosteroid (2) ^	
Low (%)	17 (15.1)
Medium (%)	44 (38.9)
High (%)	52 (46.0)
Non-adherent (%)	50.8
(Inhaler Adherence Scale ≤ 2)	

^ *n* = 13 missing due to incomplete medication strength or daily number of inhalations.

**Table 2 jpm-10-00126-t002:** Percentage of group non-adherent and adherent and risk of demographic characteristics.

	Non-Adherent *n* = 64	Adherent *n* = 62	*p* Value	Unadjusted OR * (95%CI)
*Recruitment source*			0.15	
Asthma Centre	36 (58.1% ^#^)	26 (41.9%)		1.78 (0.88, 3.61)
Volunteer	28 (43.8%)	36 (56.2%)		1.00
*Mean age (SD)*	42.8 (15.5)	55.4 (13.5)	0.0001	
*Gender*			0.07	
Male	35 (60.3%)	23 (39.7%)		2.05 (1.00, 4.17)
Female	29 (42.7%)	39 (57.3%)		1.00

* for predicting non-adherence (OR = 1.0 indicates reference group). ^#^ all (%) represent row percent.

**Table 3 jpm-10-00126-t003:** Percentage of group non-adherent and adherent and risk of asthma characteristics.

	Non-Adherent *n* = 64	Adherent *n* = 62	*p* Value	Unadjusted OR * (95%CI)
Median age at asthma diagnosis (IQR)	9.5 (27)	18.5 (40)	0.11	
Has asthma action plan	50 (50.0% ^#^)	50 (50.0%)	0.89	0.86 (0.36, 2.04)
No asthma action plan	14 (53.9%)	12 (46.1%)		1.00
Has action plan written on paper	22 (50.0%)	22 (50.0%)	1.00	0.95 (0.45, 1.98)
No written action plan	42 (53.9%)	40 (46.1%)		1.00
Had asthma education	17 (44.7%)	21 (55.3%)	0.48	0.71 (0.33, 1.52)
Never had asthma education	47 (53.4%)	41 (46.6%)		1.00
Previous hospital admission	46 (52.9%)	41 (47.1%)	0.61	1.31 (0.62, 2.79)
No previous hospital admission	18 (46.2%)	21 (53.8%)		1.00
Hospital admission in the last year	15 (71.4%)	6 (28.6%)	0.07	2.86 (1.03, 7.93)
No hospital admission in the last year	49 (46.7%)	56 (53.3%)		1.00
Reported slow onset attack before	36 (45.6%)	43 (54.4%)	0.18	0.57 (0.27, 1.18)
No slow onset attack before	28 (59.6%)	19 (40.4%)		1.00
Reported fast onset attack before	30 (50.8%)	29 (49.1%)	1.00	1.00 (0.50, 2.00)
No fast onset attack before	34 (50.7%)	33 (49.3%)		1.00
Daily dose of inhaled corticosteroid				
High	24 (46.2%)	28 (53.8%)	0.29	0.47 (0.15, 1.45)
Medium	20 (45.5%)	24 (54.5%)		0.45 (0.14, 1.45)
Low	11 (64.7%)	6 (35.3%)		1.00
Missing	9	4		
Once daily inhaled corticosteroid	5 (55.6%)	4 (44.4%)	1.00	1.23 (0.31, 4.80)
Twice daily inhaled corticosteroid	59 (50.4%)	58 (49.6%)		1.00
Wheeze—daily	13(48.2%)	14 (51.8%)	0.55 ^^^	1.12 (0.27, 4.55)
Wheeze—more than once per month	23 (62.2%)	14 (37.8%)		1.97 (0.51, 7.68)
Wheeze—less than once per month	15 (44.1%)	19 (55.9%)		0.95 (0.24, 3.71)
Not at all	5 (45.5%)	6 (54.5%)		1.00
Missing	8	9		

* for predicting non-adherence (OR = 1.0 indicates reference group). ^#^ all (%) represent row percent. ^^^ Cochrane Armitage test for trend.

**Table 4 jpm-10-00126-t004:** Percentage of group non-adherent and adherent and risk of anxiety and depression and personality domain.

	Non-Adherent *n* = 64	Adherent *n* = 62	*p* Value	Unadjusted OR * (95%CI)
*Anxiety*			0.60 ^^^	
Moderate/Severe	13 (52.0% ^#^)	12 (48.0%)		1.18 (0.47, 2.94)
Mild	17 (56.7%)	13 (43.3%)		1.42 (0.60, 3.36)
Normal	34 (47.9%)	37 (52.1%)		1.00
*Depression*			0.08 ^^^	
Moderate/Severe	2 (28.6%)	5 (71.4%)		0.34 (0.06, 1.82)
Mild	4 (33.3%)	8 (66.7%)		0.42 (0.12, 1.49)
Normal	58 (54.2%)	49 (45.8%)		1.00
*Neuroticism*			0.003 ^^^	
High	29 (67.4% ^#^)	14 (32.6%)		3.75 (1.55, 9.08)
Average	19 (50.0%)	19 (50.0%)		1.81 (0.75, 4.37)
Low	16 (35.6%)	29 (64.4%)		1.00
*Extraversion*			0.30 ^^^	
High	20 (55.6%)	16 (44.4%)		1.58 (0.65, 3.85)
Average	25 (53.2%)	22 (46.8%)		1.44 (0.63, 3.30)
Low	19 (44.2%)	24 (55.8%)		1.00
*Openness*			0.20 ^^^	
High	27 (58.7%)	19 (41.3%)		1.74 (0.74, 4.09)
Average	19 (47.5%)	21 (52.5%)		1.11 (0.46, 2.66)
Low	18 (45.0%)	22 (55.0%)		1.00
*Conscientiousness*			0.22 ^^^	
High	13 (40.6%)	19 (59.4%)		0.55 (0.22, 1.37)
Average	25 (53.2%)	22 (46.8%)		0.92 (0.41, 2.07)
Low	26 (55.3%)	21 (44.7%)		1.00
*Agreeableness*			0.44 ^^^	
High	22 (45.8%)	26 (54.2%)		0.73 (0.31, 1.69)
Average	21 (53.8%)	18 (46.2%)		1.00 (0.41, 2.44)
Low	21 (53.8%)	18 (46.2%)		1.00

* for predicting non-adherence (OR = 1.0 indicates reference group). ^#^ all (%) represent row percent. ^^^ Cochrane Armitage test for trend.

**Table 5 jpm-10-00126-t005:** Median scores for asthma practical knowledge and perceived involvement in care scales, and mean optimism for adherent and non-adherent groups.

	Non-Adherent *n* = 64	Adherent *n* = 62	*p* Value
*Asthma practical knowledge*			
Median slow onset attack score (IQR)	10.0 (7.5)	12.5 (6.0)	0.55
Median fast onset attack score (IQR)	11.0 (12.0)	12.0 (8.0)	0.21
*Perceived Involvement in Care Scale*			
Overall median PICS score (IQR)	19.0 (3.5)	20.0 (4.0)	0.33
Median Doctor Facilitation score (IQR)	8.0 (3.0)	9.0 (3.0)	0.21
Median Patient Information score (IQR)	6.0 (2.0)	7.0 (3.0)	0.87
Median Patient Decision Making score (IQR)	5.0 (2.0)	5.0 (2.0)	0.63
*Optimism*			
Mean Optimism score (SD)	−0.05 (4.2)	−0.6 (4.7)	0.45

**Table 6 jpm-10-00126-t006:** Predictors of non-adherence to inhaled corticosteroid therapy identified in a multi-variate model.

	Unadjusted OR * (95% CI)	Adjusted OR * (95% CI)	*p* Value
Each 10 year increase in age		0.94 (0.91, 0.97)	0.001
*Neuroticism*			
High	3.75 (1.55, 9.08)	5.71 (1.79, 18.29)	0.003
Average	1.81 (0.75, 4.37)	1.64 (0.58, 4.62)	0.35
Low	1.00	1.00	
*Depression*			
Moderate/Severe	0.34 (0.06, 1.82)	0.08 (0.01, 0.76)	0.03
Mild	0.42 (0.12, 1.49)	0.15 (0.03, 0.80)	0.03
Normal	1.00	1.00	
Hospital admission within the last year	2.86 (1.03, 7.93)	4.60 (1.25, 16.9)	0.02
No hospital admission within the last year	1.00		
*Gender*			
Male	2.05 (1.00, 4.17)	1.97 (0.81, 4.77)	0.13
Female	1.00	1.00	

* for predicting non-adherence (OR = 1.0 indicates reference group, except for age where OR relates to effect of a ten year increase in age).

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
