# Peer review of "Psychological and Medical Characteristics Associated with Non-Adherence to Prescribed Daily Inhaled Corticosteroid"

_jpm, 2020, doi:10.3390/jpm10030126_

Round 1
Reviewer 1 Report
Toelle Brett G et.al analyzed various medical and individual psychological characteristics potentially associated with non-observance to the main inhaled treatment employed in asthma.
As introduced by authors, non-adherence remain a very common problematic in this chronic pulmonary disease and identification of factors associated with inobservance could lead to identify population benefiting from personalized/educational learning sessions.
In general, the manuscript is well organized, with scientific relevance in asthma field and well discussed.
I would only suggest some minor points:
- please check format / presentation for all tables (many sentences are not readable)
- table 5 : please check and correct "P-valu"
- If available, authors could present and analyze association between non-observance and
- others asthma related treatments (beta 2 inhaled treatment, biotherapies...)
- asthma severity
- asthma control (assessed by ACQ?)
- smoking history
- If not available, it might be mentionned in discussion section
Author Response
Toelle Brett G et.al analyzed various medical and individual psychological characteristics potentially associated with non-observance to the main inhaled treatment employed in asthma.
As introduced by authors, non-adherence remain a very common problematic in this chronic pulmonary disease and identification of factors associated with inobservance could lead to identify population benefiting from personalized/educational learning sessions.
In general, the manuscript is well organized, with scientific relevance in asthma field and well discussed.
Thank you.
I would only suggest some minor points:
please check format / presentation for all tables (many sentences are not readable)
We apologize for the problems with the formatting and presentation of all the tables. This appears to have been the result of conversion of the submitted MS Word document to the PDF file. We will upload both an MS Word version of the manuscript and a PDF version of the manuscript to ensure the formatting for all tables is readable.
table 5 : please check and correct "P-valu"
Sorry again. On the MS Word version this is correct but it got cut off in the conversion to PDF.
If available, authors could present and analyze association between non-observance and
others asthma related treatments (beta 2 inhaled treatment, biotherapies...)
We hope that our correction of the formatting of Table 3 will show that we had conducted and included analyses that will address this request for additional analyses.
All participants were prescribed as needed beta 2 inhaled treatment and so this is not a characteristic that is likely to differentiate between adherent and non-adherent participants. We did not ask about biotherapies. However, at the bottom of Table 3 we assessed the association with once daily or twice daily inhaled corticosteroid medication and we showed that neither once daily nor twice daily ICS medication was associated with non-adherence.
asthma severity
Table 3 contains some asthma characteristics to assess the association of asthma severity and non-adherence. We assessed hospitalization within the last 12 months and showed that this characteristic was associated with non-adherence. We also assessed frequency of wheeze and showed that those who had experienced the daily symptom of wheeze were just as likely to be non-adherent as those who experienced wheeze less than once per month.
asthma control (assessed by ACQ?)
We did not collect information on asthma control assessed by the ACQ.
smoking history
We did not collect information on smoking history.
If not available, it might be mentioned in discussion section
Ln 284-287 we added this additional text “We did not collect information about asthma control or smoking history so we are not able to examine the association of these characteristics with non-adherence which would have been quite interesting.”
Reviewer 2 Report
The manuscript entitled “Psychological and medical characteristics associated with non-adherence to prescribed daily inhaled corticosteroid” describes relationship between psychological characteristics and adherence to inhaled corticosteroid (ICS).
The investigation into the relationship between the several psychological aspects and adherence to ICS is interesting and the number of patients is enough.
However, most of the treating physicians are interested in the outcome of each relationship, in other words, the psychological patterns which are related to poor adherence to ICS, leading to the poor control and poor future risk for exacerbation or hospitalization.
Apart from the psychological aspect, the treatment strength is considered to affect adherence to asthma treatment. Therefore, the dose of ICS as well as ICS alone, ICS/LABA (long-acting β2-agonist) or more should be described. In particular, formoterol (FM) is the LABA with immediate response like short-acting β2-agonist (SABA), BUD/FM could show better adherence, which should be considered.
In the discussion (line 341), the authors referred to the adherence in diabetes treatment. It is not appropriate to compare the adherence in asymptomatic disease like diabetes.
As to the psychological aspect, the authors have mentioned on the difference between the observed results and the previous reports with depression. I want to know what we can learn from the current study which will improve the adherence in the treatment of asthma.
Author Response
The manuscript entitled “Psychological and medical characteristics associated with non-adherence to prescribed daily inhaled corticosteroid” describes relationship between psychological characteristics and adherence to inhaled corticosteroid (ICS).
The investigation into the relationship between the several psychological aspects and adherence to ICS is interesting and the number of patients is enough.
Thank you.
However, most of the treating physicians are interested in the outcome of each relationship, in other words, the psychological patterns which are related to poor adherence to ICS, leading to the poor control and poor future risk for exacerbation or hospitalization.
We agree and we have examined a range of psychological and medical characteristics to see how useful they are to treating physicians. Treating physicians must rely on instinct and clinical experience to make this determination, we were hoping to provide them with additional validated tools to provide greater precision around their capacity to predict “poor adherence to ICS, leading to poor control and poor future risk for exacerbation or hospitalization”.
Ln 356 to 363 we have discussed the difficulty that clinician have when predicting which of their patients will be non-adherent with treatment and then at risk of negative consequence. We hope this discussion paragraph reassures the reviewer that we agree with their point.
Apart from the psychological aspect, the treatment strength is considered to affect adherence to asthma treatment. Therefore, the dose of ICS as well as ICS alone, ICS/LABA (long-acting β2-agonist) or more should be described. In particular, formoterol (FM) is the LABA with
immediate response like short-acting β2-agonist (SABA), BUD/FM could show better adherence, which should be considered.
The reviewer raises an important point about the role of medication in adherence. Numerous previously published studies have reported on non-adherence related to medication, typically frequency of dosing. Because of these findings we did not seek to replicate those analyses in this report.
However, it is very important for us to point out that participants in this study were not using combination therapy and so we report on adherence with ICS only. We thank the reviewer for pointing this out and we have added text that explicitly states that participants in this study were not prescribed combination therapy in a single inhaler and so our measure of adherence is to the inhaled steroid only. Ln 109, we have added “(not combination)”
We have also added text to the discussion to address this important point at Ln 284-287, “We also restricted participants to those who used an ICS inhaler because measuring adherence with a combination inhaler would be influenced by the symptomatic relief provided by the bronchodilator.”
In the discussion (line 341), the authors referred to the adherence in diabetes treatment. It is not appropriate to compare the adherence in asymptomatic disease like diabetes.
We downloaded the latest version of the Word document from the website. However, the Word version did not have line numbering. We inserted line numbering but this did not match with the line numbering referred to by the reviewers. Ln 341 reference by the reviewer is Ln 317 in the revised manuscript.
We understand the concern of the reviewer and have removed this sentence and reference from the manuscript. Removed “studies in other conditions have investigated optimism with varying results. In a study of type II diabetic patients, multivariate analyses showed that pessimism was associated with adherence (28).”
As to the psychological aspect, the authors have mentioned on the difference between the observed results and the previous reports with depression. I want to know what we can learn from the current study which will improve the adherence in the treatment of asthma.
We thank the reviewer and share their interest in the unexpected finding that depression was associated with non-adherence. In the discussion we have added the following text Ln 371-375 “One possibility for this unexpected association is that participants who were depressed took their anti-depressant medication and this acted as a prompt to take their ICS medication. Unfortunately, we do not have information about other concomitant medications and so this hypothesis is speculation, but worthy of investigation in future studies.”
BMC Psychology has now accepted our accompanying manuscript that describes the validation of the adherence measure used in this report so we have placed this at Ln 137 and Ln 278 and in the reference list.
Toelle BG, Marks GB, Dunn SM. Validation of the Inhaler Adherence Questionnaire.
BMC Psychology. 2020. (Accepted 14/08/2020).
Thank you for the opportunity to revise our manuscript and we hope that it is now in a form suitable for publication.
Round 2
Reviewer 2 Report
The article has been well revised except for the effect of the evaluation of psychological and medical characteristics on the asthma treatment outcome.
The authors responded to the reviewer’s report that the subjects were treated with only inhaled corticosteroid (ICS) without further combination therapy. The added sentences in the revised manuscript are well written and understandable. However, those asthma patients who are controllable with ICS only should be stable population. There is a controversy these days on the necessity of ICS in very light asthma patients. Therefore, the subjects in the current study could be patients who could do well without ICS for at least a couple of months without exacerbation.
Thus, stratifying the subject according to the dose of ICS, which means the strength of treatment, is important in this study. Although there are numerous studies on non-adherence related to medication as the authors responded, integrating the results obtained in the current study with the medical aspect is very important, which will enrich the quality of the current manuscript.
Author Response
The article has been well revised except for the effect of the evaluation of psychological and medical characteristics on the asthma treatment outcome.
The authors responded to the reviewer’s report that the subjects were treated with only inhaled corticosteroid (ICS) without further combination therapy. The added sentences in the revised manuscript are well written and understandable.
Unfortunately, the sentence we added was incomplete. At Ln 109 It reads “(not combination) however it should state “(not combination including formoterol).” As the reviewer pointed out in their earlier review the combination inhaler that include formoterol is different because the long acting beta agonist (LABA) formoterol has a fast action of onset and has also been widely promoted as both a maintenance and reliever therapy (SMART). Therefore, it is not possible to differentiate between the use of this medication for maintenance (like ICS) or relief of symptoms.
We have also corrected the sentences at Ln 280-283 where we have added the underlined words “We also restricted participants to those who used an ICS inhaler, not in combination with formoterol, because measuring adherence with a combination inhaler containing formoterol would be influenced by the rapid onset of symptomatic relief provided by the bronchodilator.”
To assist the reader and to address the issues raised by the reviewer we have made a number of additions to the manuscript. In Table 1 “Characteristics of the 126 participants who completed the study” we have included the proportion using each type of prescribed inhaled corticosteroid medication. This provides further clarity about the types of participants included in this study. We did enrol participants who used combination therapy, but the fluticasone/salmeterol combination therapy does not provide immediate symptomatic relief and is not indicated for this purpose.
In Table 1, we have also included the daily dose of prescribed inhaled corticosteroid according to Global Initiative for Asthma guidelines which assigns doses as mild, moderate and high (1).
However, those asthma patients who are controllable with ICS only should be stable population. There is a controversy these days on the necessity of ICS in very light asthma
patients. Therefore, the subjects in the current study could be patients who could do well without ICS for at least a couple of months without exacerbation.
We apologise for including an incomplete sentence that inadvertently created an impression that the participants in this study were mild, were managed on ICS only and “who could do well with ICS for at least a couple of months without exacerbation”. As we now show in Table 1 there were 44.4% who were prescribed a combination ICS/LABA medication and 46.0% were prescribed a high dose of inhaled corticosteroid medication.
We thank the reviewer for raising these questions and hope that the inclusion of these two additional pieces of data will assist the reader to understand the type of participants included in this study.
Thus, stratifying the subject according to the dose of ICS, which means the strength of treatment, is important in this study. Although there are numerous studies on non-adherence related to medication as the authors responded, integrating the results obtained in the current study with the medical aspect is very important, which will enrich the quality of the current manuscript.
We accept that it may be of interest to stratify the analyses according to the strength of treatment, however we have not done this for two reasons.
Firstly, we have analysed the association between daily inhaled steroid dose and adherence and have shown that there was no statistically significant association. That is participants in this study who were prescribed low, medium or high doses of inhaled corticosteroid were just as likely to be non-adherent with their ICS medication.
Secondly, we did not power this study to facilitate a stratified analyses. Overall we had 126 participants with 15% of the sample in the low ICS group, 39% in the medium group and 46% in the high prescribed ICS. Therefore, the smaller numbers in each stratified group will provide reduced precision and the results from this unplanned additional analyses will not be reliable.
We have added this additional text at Ln 287-291 to the discussion. “In this study we did not find an association between daily corticosteroid dose and adherence but a larger study would be sufficiently powered to perform a stratified analyses of the association of these psychological and medical characteristics with adherence for different levels of daily corticosteroid dose.”
Thank you for the opportunity to revise our manuscript and we hope that it is now in a form suitable for publication.